# Telomere Maintenance Mechanisms in a Cohort of High-Risk Neuroblastoma Tumors and Its Relation to Genomic Variants in the *TERT* and *ATRX* Genes

**DOI:** 10.3390/cancers15245732

**Published:** 2023-12-07

**Authors:** Anna Djos, Ketan Thombare, Roshan Vaid, Jennie Gaarder, Ganesh Umapathy, Susanne E. Reinsbach, Kleopatra Georgantzi, Jakob Stenman, Helena Carén, Torben Ek, Tanmoy Mondal, Per Kogner, Tommy Martinsson, Susanne Fransson

**Affiliations:** 1Department of Laboratory Medicine, Institute of Biomedicine, Sahlgrenska Academy, University of Gothenburg, 40530 Gothenburg, Sweden; anna.djos@gu.se (A.D.); ketan.thombare@gu.se (K.T.); roshan.vaid@medkem.gu.se (R.V.); jennie.gaarder@gu.se (J.G.); ganesh.umapathy@gu.se (G.U.); tanmoy.mondal@medkem.gu.se (T.M.); tommy.martinsson@gu.se (T.M.); 2Department of Clinical Genetics and Genomics, Sahlgrenska University Hospital, 41345 Gothenburg, Sweden; 3Department of Medical Biochemistry and Cell Biology, Institute of Biomedicine, Sahlgrenska Academy, University of Gothenburg, 40530 Gothenburg, Sweden; susanne.reinsbach@scilifelab.se; 4Childhood Cancer Research Unit, Department of Women’s and Children’s Health, Karolinska Institutet, 17177 Stockholm, Sweden; kleopatra.georgantzi@regionstockholm.se (K.G.); jakob.stenman@regionstockholm.se (J.S.); per.kogner@ki.se (P.K.); 5Sahlgrenska Center for Cancer Research, Department of Medical Biochemistry and Cell Biology, Institute of Biomedicine, Sahlgrenska Academy, University of Gothenburg, 40530 Gothenburg, Sweden; helena.caren@gu.se; 6Children’s Cancer Center, Sahlgrenska University Hospital, 41650 Gothenburg, Sweden; torben.ek@vgregion.se; 7Department of Clinical Chemistry, Sahlgrenska University Hospital, 41345 Gothenburg, Sweden

**Keywords:** neuroblastoma, telomeres, TMM, *TERT*, *ATRX*, ALT, pediatric cancer

## Abstract

**Simple Summary:**

Immortalization is a hallmark of malignant tumors, including in pediatric cancer neuroblastoma, where it is associated with an adverse prognosis. In this study, we characterized a Swedish neuroblastoma cohort with the focus on telomere maintenance mechanisms (TMMs), i.e., *MYCN* amplification and the juxtapositioning of *TERT* and *ATRX* aberrations. We show a strong correlation between *ATRX* aberrations and ALT and that aberrations affecting *ATRX* or *TERT* are enriched in high-risk cases with 11q deletion. This study, thereby, supports the need for further advancement of TMM-targeted therapies for this subgroup of neuroblastoma patients.

**Abstract:**

Tumor cells are hallmarked by their capacity to undergo unlimited cell divisions, commonly accomplished either by mechanisms that activate *TERT* or through the alternative lengthening of telomeres pathway. Neuroblastoma is a heterogeneous pediatric cancer, and the aim of this study was to characterize telomere maintenance mechanisms in a high-risk neuroblastoma cohort. All tumor samples were profiled with SNP microarrays and, when material was available, subjected to whole genome sequencing (WGS). Telomere length was estimated from WGS data, samples were assayed for the ALT biomarker c-circles, and selected samples were subjected to methylation array analysis. Samples with *ATRX* aberration in this study were positive for c-circles, whereas samples with either *MYCN* amplification or *TERT* re-arrangement were negative for c-circles. Both *ATRX* aberrations and *TERT* re-arrangement were enriched in 11q-deleted samples. An association between older age at diagnosis and 1q-deletion was found in the ALT-positive group. *TERT* was frequently placed in juxtaposition to a previously established gene in neuroblastoma tumorigenesis or cancer in general. Given the importance of high-risk neuroblastoma, means for mitigating active telomere maintenance must be therapeutically explored.

## 1. Introduction

The telomeres are essential structures that cap and protect the integrity of chromosomes, thereby ensuring that the chromosome ends will not be recognized as double-strand DNA breaks. The telomeres also prevent loss of coding DNA due to replication as each cell division will lead to a shortening of the telomeres with approximately 50 base pairs [1,2]. This gradual telomere erosion provides an inherent limitation of replication, and normal cells will halt cell cycle progression and enter senescence when the telomers reach a critical threshold. Tumor cells are, on the other hand, hallmarked by the capacity to undergo unlimited cell divisions. They must, therefore, adopt different telomere maintenance mechanisms (TMMs) in order to maintain telomere length and avoid replication crisis. This is commonly accomplished either by mechanisms that activate *TERT* or through the alternative lengthening of telomeres (ALT) pathway. Activation of *TERT* that leads to expression of the telomerase enzyme is seen in ~85% of all human tumors and could be achieved through promoter mutation, structural rearrangement, amplification, or transcription factor-driven re-expression. ALT is a telomerase-independent TMM caused by a homologous recombination process with implication of the break-induced repair machinery. ALT is associated with extensive telomeres together with the accumulation of extrachromosomal telomeric circles (c-circles), the presence of ALT-associated PML nuclear bodies (APBs), and they typically show loss of full-length ATRX and, more rarely, DAXX [3].

Neuroblastoma (NB) is a pediatric cancer derived from immature cells of the sympathetic nervous system, with highly heterogeneous clinical and biological behavior. Tumors with good prognosis are generally near the triploid and display aneuploidy with whole chromosome gains and losses, whereas high-risk tumors with poor prognosis are near the di- or tetraploid and are associated with recurrent segmental chromosomal aberrations (SCAs), such as deletion of chromosome arm 11q or amplification of the *MYCN* oncogene (MNA) [4]. It has also been observed that the presence of TMM is highly associated with poor prognosis, regardless of NB risk group or stage [5]. In NB tumors with MNA, telomere maintenance is commonly achieved through *MYCN*-driven re-expression of *TERT*, whereas the non-MNA high-risk NB cases use other means for telomere maintenance. In this latter group, genomic alterations of *TERT* or *ATRX* are common [5,6]. *TERT* promotor mutations are highly recurrent in some malignancies but have not been reported in NB [7]. More common are genomic re-arrangements that position *TERT* under the control of an active promotor (super-enhancer hijacking), leading to telomerase expression. In addition to structural variants of the *TERT* locus, multi-exon deletions and loss-of-function mutations in *ATRX* are also frequently seen in non-MNA high-risk NBs. Inactivating mutations and deletions of *ATRX* are associated with long telomeres due to ALT [8]. ALT-positive tumor cells show upregulation of the transcription of TERRA, a long non-coding RNA that is thought to contribute to the maintenance of long telomeres [9,10]. A recent study showed that METTL3-mediated N6-methyladenosine (m^6^A) modification of TERRA is essential for maintaining long telomeres in ALT cells [11]. Thus, inhibition of METTL3 could be a therapeutic option for ALT-positive NB [12].

However, only 55–60% of ALT-positive NBs are reported to have *ATRX* aberrations [13,14,15,16], indicating other means of achieving ALT. Similar numbers have been reported in other cancer forms [17,18,19,20] and, furthermore, several studies indicate that *ATRX* loss alone is not sufficient for the development of ALT [21,22]. Hartlieb et al. reported an association between 1q-deletion and ALT, indicating there could be an ALT suppressor located in this region [15]. In addition to the reactivation of *TERT* expression and ALT, one study indicated the presence of a variant TMM that lacks both telomerase activity and ALT, so-called ever-shortening telomeres (ESTs), in a NB subgroup [13]. Despite the lack of ALT, this group still had expanded telomeres, and the authors hypothesized this to be caused by failure to form T-loops that prevent over-lengthening of the telomeres during embryogenesis when the telomerase is active normally [13].

## 2. Materials and Methods

### 2.1. Sample Collection and Patients

Tumor tissue was collected after written or verbal informed consent was obtained by parents/guardians. Genomic DNA was extracted from fresh frozen tumor or blood using the DNeasy Blood and Tissue kit (Qiagen, Hilden, Germany) according to the manufacturer’s protocol. In total, 232 NB tumor samples were screened with SNP microarray. Of these 232 tumors, 50 were subjected to whole genome sequencing, and 28 were screened for *ATRX* aberrations with MLPA. In total, 21 with *ATRX* aberration and 23 with *TERT* re-arrangement were detected (n = 44). These 44 patients together with 11 MNA patients, 13 patients with 11q-deleted profile with no detected alteration in either *ATRX*, *TERT*, or *MYCN*, 3 with other structural profiles, and 6 with numerical only profile made up the study cohort of 77 patients subjected to extended analyses (Table 1). Clinical and biological characteristics of tumor samples in this cohort are listed in Table 1. The NB cell lines SK-N-BE(2) and SK-N-FI were obtained from the American Type Culture Collection and cultured as previously described [23].

### 2.2. SNP Microarray Analysis

NB tumor samples (n = 232) were screened for copy-number variants and allelic imbalances using Affymetrix CytoScan HD SNP microarrays (Thermo Fisher Scientific, Waltham, MA, USA), as described earlier [24]. For primary data analysis, GeneChip Command Console Software version 5.0.0.368 (Thermo Fisher Scientific) was used, while genomic profiles and amplicon boundaries were determined using Chromosome Analysis Suite (ChAS v.3.3.0; Thermo Fisher Scientific).

### 2.3. Whole Genome Sequencing

Whole genome sequencing (WGS) was performed on DNA from tumor material and matched constitutional DNA from 50 unique patients previously analyzed with SNP microarray. This DNA was subjected to sequencing and bioinformatical handling, as described previously [25]. Briefly, sequencing was performed on Illumina instrumentation (Illumina, San Diego, CA, USA) at Clinical Genomics, SciLife Laboratories, Gothenburg/Stockholm, Sweden. Mapping, realignment around indels, and variant calling were carried out using the Sentieon suite of bioinformatics tools (Sentieon Inc., Mountain View, CA, USA) (Sentieon version v.201808.03). Only high-quality called SNVs with a minimum of 10% variant allele frequency and a total read coverage of ten were considered for further analysis and evaluation. All synonymous variants or variants in non-coding regions were excluded, except those affecting canonical splice sites. The Canvas tool (version 1.38.0.1554) [26] was used to call copy-number alterations while structural variants were called using the Manta tool (version 1.1.1) [27] with filtering based on germline variation, artifacts caused by problematic regions, or presence in the SweGen Variant Frequency dataset (https://swefreq.nbis.se/, accessed on 18 November 2022) or in our in-house set of normal controls. Patient-specific origin of tumor-normal pairs was verified using a previously developed Python script that calculates the shared fraction from 400,000 SNPs [25]. Calculation of telomere length was performed using the TelSeq software v.1.0 that estimates telomere length, as defined as repeats of more than seven TTAGGG motifs from whole genome sequencing data [28]. Telomere length in tumors was normalized against patients’ telomere length in blood lymphocytes. Distance between break points at the *TERT* loci and super-enhancer (SE) elements was estimated using previously reported SE positions, as noted in the studies by van Groningen et al. 2017 [29] and Gartlgruber et al. 2021 [30].

### 2.4. MLPA Multiplex Ligation-Dependent Probe Amplification (MLPA)

Genomic DNA from 28 NB patients, with age of onset above 72 months, was analyzed for *ATRX* deletions and duplications using SALSA MLPA probe mix P013-A1 ATRX according to the manufacturer’s instructions, version MDP-v002 (MRC Holland, Amsterdam, The Netherlands). Lymphocyte DNA from healthy blood donors was used as controls. The probe mix covers all *ATRX* exons as well as nine genomic reference probes. Briefly, 200 ng of genomic DNA was denatured prior to overnight hybridization with the MLPA probes, ligation, and PCR amplification probe ligation products. Separation of amplification was carried out on a 3730 DNA Analyzer (Applied Biosystems, Foster City, CA, USA), and raw data were analyzed using Gene Mapper Software 4.0 (Applied Biosystems).

### 2.5. C-Circle Assays

The C-circle assay technique was described by Henson et al. [31] and detects partially single-stranded telomeric (CCCTAA)n DNA circles, and it is widely used to assay the ALT biomarker C-circles. In total, 79 NB samples of different genomic subgroups from 72 patients were analyzed using C-circle assay. Briefly, genomic DNA (80 ng) was digested with the restriction enzymes 4 U/μg HinfI and 4 U/μg RsaI and 25 ng/μg of RNase. In the rolling amplification reaction (RCA), 40 ng (12.5 μL) of digested DNA of each sample was combined with 12.5μL master mix of 0.2 mg/mL BSA, 0.1% Tween, 1 mM each dATP, dGTP, and dTTP, 2X Phi29 buffer, and 7.5 U Phi29 DNA polymerase (Phi29+) and incubated at 30 °C for 8 h, then at 65 °C for 20 min. In order to correct for sample-specific background, a second reaction for all samples was run in parallel with Phi29 DNA polymerase omitted (Phi29−). On every blot, the NB cell line SK-N-FI was included as an ALT-positive control, SK-N-BE(2) as an ALT-negative control, and a control with DNA omitted, non-template control (NTC). The RCA products were diluted to 200 µL with nuclease-free water and slot-blotted to a 6XSSC-soaked Hybond N+ nylon membrane (GE Healthcare, Chicago, IL, USA). Slot-blotted RCA products were UV cross-linked to the membrane and hybridized at 37 °C with DIG-labelled telomere probe (MERCK) in Dig Easy hyb buffer (Roche Diagnostics, Basel, Switzerland) overnight. The membrane was then subjected to washing and blocking using the Dig wash and block buffer set (Roche), before being incubated with anti-DIG antibody conjugated with alkaline phosphatase (Roche). The membrane was further washed and incubated in dark with the chemiluminescent substrate CDP-star before being exposed to high-resolution chemiluminescence using ChemiDoc Imaging System version 2.4.0.03 (Bio-Rad Laboratories, Hercules, CA, USA). The band intensities were measured with Image lab software v6.1 (Bio-Rad Laboratories) and signal intensity ratios of Phi29+ control versus Phi29− control was calculated. The ratios were also normalized against the ratio of the positive control SK-N-FI included in each experiment. To be scored as c-circle positive, the signal intensity had to be increased by at least 2-fold when comparing Phi29+ with Phi29− reaction and at least 40% the intensity of the SK-N-FI signal on the same blot.

### 2.6. DNA Methylation Analysis

DNA methylation analysis was performed on a small subset of the cohort (n = 8). Prior to the DNA methylation array assay, bisulfite modification was performed using the EZ DNA methylation kit (Zymo Research, Irvine, CA, USA) according to the manufacturers’ protocol. An amount of 150ng of bisulphite-modified DNA was applied to Infinium EPIC Bead Chip (Illumina) with consecutive scanning using the Illumina NextSeq 550 scanner. The resulting IDAT-files were uploaded and analyzed using CNS classifier version 12.5, available at the molecular neuropathology platform (https://www.molecularneuropathology.org/mnp/, accessed on 20 July 2023). Despite main focus on different CNS malignancies, the CNS classifier 12.5 also includes methylation classification of peripheral neuroblastoma with subclasses “Neuroblastoma, MYCN type”, “neuroblastoma, TMM negative”, and “neuroblastoma, ALT/TERT TMM positive”.

### 2.7. Kaplan–Meier Analysis

Kaplan–Meier analysis was utilized with calculation of log-rank test for overall survival probability in relation to the expression levels of selected genes on 1q (*TRIM67*, *SPRTN*, *FH*, *EXO1* and *SMYD3*), focusing on non-*MYCN*-amplified NB due to low ALT incidence in MNA tumors. The Kaplan–Meier analyses were performed using the Kaplan-scan cutoff method in ‘R2: Genomic Analysis and Visualization Platform’ (http://r2.amc.nl, accessed on 30 October 2023) with the publicly available datasets ‘Tumor Neuroblastoma-SEQC-498-rpm-Seqnb1’ (n = 498) and ‘Versteeg’ (n = 88). The Kaplan-scan cutoff method examines every increasing expression value as cutoff for log-rank test in order to find the optimal segregation point of two groups based on gene expression. This method then presents the most statistically significant cutoff with corresponding Bonferroni corrected *p*-value together with the initial non-corrected *p*-value.

## 3. Results

### 3.1. TERT Aberrations

Segmental aberrations in the proximity of *TERT* were detected through SNP microarray and/or WGS in tumor material from 23 unique patients in total. The *TERT* aberrations were commonly associated with either a focal gain with breakpoints located both proximally and distally of the *TERT* locus, or 5p-gain with break downstream of *TERT* (Figure 1A and Appendix A). WGS data for available cases showed that a majority of structural aberrations caused the juxtaposition of *TERT* to gene regions with strong enhancer elements, including NB core regulatory circuit genes, such as *HAND2*, *ISL1,* and *MYCN* (Figure 1A,B). The distance from the *TERT* transcription start site to the closest super-enhancer element was commonly less than 300 kb, with the shortest distance being 1342 bp (Appendix A). Most of the enhancer-associated regions are fused to a break point located proximally to *TERT,* but one case showed a distal break of *TERT* due to an unbalanced translocation between chromosomes 4 and 5, connecting the *TERT* and *HAND2* loci (Figure 1B, bottom panel). Recurrent *TERT* juxtaposition was seen for the loci of *DLG2*, *CCND1,* and *MYCN* (Figure 1A,B). Furthermore, no point mutation in *TERT* or the *TERT* promotor was detected in the sequenced cases.

### 3.2. ATRX Aberrations

In the investigated cohort, 22 *ATRX* aberrations were detected in tumor material from 21 patients, as judged from SNP microarrays, MLPA, and/or WGS (Table 2 and Figure 2). Among these, eighteen patients had tumors with structural variations in *ATRX,* whereof sixteen were multi-exon deletions. This includes one patient where the biopsy from the primary tumor displayed a larger inversion with a break within *ATRX,* while later material from the resected tumor showed an intragenic deletion (case 76R4/B). In two patients, *ATRX* duplications were detected, involving exon 2–9 with a breakpoint within exon 9 (case 60R6B) and exon 27–30 in case 69R8A/B. NB tumors from three different patients displayed single-nucleotide variants in *ATRX*. Case 70R2A had a somatic nonsense variant p.(R770*), case 79R2A had a somatic missense variant p.(E2246A) predicted to be damaging and possibly damaging by SIFT and PolyPhen2, respectively, while case 30R0 displayed a rare (gnomAD minor allele frequency = 0.0005841) germline missense variant p.(D937N). However, this variant has been reported as benign/likely benign in ClinVar (VCV000696966.12) in the context of Alpha-thalassemia/impaired intellectual development syndrome. Consistent with previous reports on *ATRX* aberrations [32,33], deletions of exon 2–10 (n = 4) and 2–9 (n = 2) were the two most common alterations and also the only two recurrent deletions. Based on exons included in the deletion and the possibility of exon 10 skipping, at least 10 out of 16 multiexon deletions could generate in-frame fusions. Details of the *ATRX* aberrations are listed in Table 2.

### 3.3. C-Circle Assay

Altogether, 79 NB samples of different genomic subgroups from 72 patients were analyzed using the C-circle assay. Among the 21 patients with *ATRX* aberration, material from 18 patients was available for analysis with the C-circle assay. All analyzed samples from tumors with structural *ATRX* aberration (n = 17), as well as the sample (70R2A) harboring the nonsense variant p.(R770*), scored as positive in the C-circle assay (Figure 3). Sample (30R0) with the germline variant p.D937N scored c-circle negative, in line with previously reported ClinVar interpretation of pathogenicity. Sample (79R2A) with the somatic p.(E2246A) variant indicated the presence of both c-circles and a pre-existing single-stranded telomeric G-strand, as judged by detectable signals in both Phi29+ and Phi29−. Thus, the ratio of Phi29+/Phi29− did not reach the set threshold level for c-circle positivity, despite displaying 43% of the signal quotient of SK-N-FI Phi29+/Phi29−.

Among the 24 *TERT*-rearranged tumor samples from 23 unique patients, material was available for the C-circle assay for all but one sample. All cases with *TERT*-associated structural aberrations (n = 23) scored as c-circle negative (Figure 3). Fifteen samples from eleven unique patients with *MYCN* amplification but without detected alterations in *ATRX* or *TERT* were examined using the C-circle assay, and all samples (n = 15) scored as c-circle negative. Twenty-one samples with no detected alteration in *ATRX*, *TERT*, or *MYCN*, including twelve with 11q-deleted profiles, three with other structural profiles and six with numerical-only profiles, also scored as negative for the presence of c-circles. However, four samples (66R3C, 32R2B, 73R5D, and 59R7C) showed signal in both Phi29+ and Phi29− reactions (Figure 3). The uncropped blots are presented in Appendix A. 

### 3.4. Methylation Array Profiling

All investigated cases received a calibrated score >0.99 for the methylation family “neuroblastoma”, with the results from representative cases of different genomic NB subgroups shown in Figure 4A. While the sample with a numerical-only profile was scored as subclass “TMM-negative”, all samples (n = 3) with *ATRX* aberrations received the highest score for subclass ALT/TERT TMM. This included case 79R2A with a somatic p.E2246A *ATRX* variant and ambiguous c-circle positivity due to signal in both Phi29+and Phi29− that received a calibrated score of 0.98 for ALT/TERT TMM (Figure 4B, upper panel). Two of three cases with MNA received the highest scores for subclass “MYCN type”. The three investigated MNA cases included one case (16R4) with ATRX deletion (exon 20--25) and one case (73R3A) with a *TERT* SV (breakpoint 1.1 kb downstream of *TERT* and fusion to the *MYCN* locus). The case with co-occurring MNA and *ATRX* with the presence of c-circles received a score below a confident prediction of >0.9 (score = 0.69) for ALT/TERT TMM, while the score for subclass “MYCN type” was 0.29 (Figure 4B, second upper panel). The case with co-occurring MNA and *TERT* was confidently classified to “MYCN type” (score = 0.99) (Figure 4B, second bottom panel). Interestingly, the 11q-deleted sample with sample 66R3C with a c-circle signal in both Phi29+ and Phi29− reactions received a confident score of 0.97 for subclass ALT/TERT TMM.

### 3.5. Telomere Length Estimation

A significant increase in normalized telomere length was seen among samples with *ATRX* aberrations (average ratio 1.45), as compared to samples with a *TERT* alteration (average ratio 0.93) or the entire group of samples lacking detectable *ATRX* aberration (*p* = 0.02 and *p* = 0.05, respectively, Student’s two-sided t-test; Figure 5A). The corresponding normalized telomere ratios in samples with MNA, including those co-occurring with 11q-del, were 0.92, in 11q-deleted samples 1.54, in the other segmental group 1.00, in the 17q-gain group 2.19, and in the numerical-only group 0.81. Indication of long tumor telomeres in the absence of *ATRX* aberrations was noted for cases 32R2B and 66R3A/C (Figure 5A), while case 77R0A presented long telomeres in both tumor and corresponding normal (Appendix A).

### 3.6. Correlation with Age and Segmental Aberrations on Chromosome 1, 11, and MYCN

Segmental aberrations at chromosome 1 were determined from SNPs with 1p/1q-deletion or 1q-gain, defined as a reduced or gained copy number compared to the neighboring segment. Among the complete cohort, 17 cases had a 1q-gain, 32 cases had a 1p-deletion, and 13 cases had 1q-deletion (Figure 5B). 1p-deletion was present in all MNA tumors but was also detected in the *TERT* SV group and, more rarely, in the group with *ATRX* aberration. The 1q-deletion-associated breakpoints were located within a hot-spot region at chr1q42.2 between the genes *FAM89A* and *TRIM67* and were almost exclusively found in tumors with *ATRX* aberrations (92%, 12/13). 1q-gain was present in 17 cases, predominantly in samples with *TERT* SV (64.7%, 11/17) and, to a lesser extent, among samples with *ATRX* aberration (17.6% 3/17). The 1q-gain breakpoints were scattered at different locations on chromosome 1.

11q-deletion was present in the majority of cases with *ATRX* and *TERT*; 14 out of 22 samples (64%) with *ATRX* aberration and 17 of 23 cases (74%) with *TERT* SV displayed 11q-deletion. The co-occurrence of MNA and *ATRX* deletion was only present in one single case (16R4) (Figure 4B, upper panel), while simultaneous occurrences of MNA and *TERT* SV were present in five cases, although 11q-deletion was also detected in three of these (Table 1).

The average age of onset was significantly higher among cases with *ATRX* aberration (97 months), as compared to cases with *TERT* SV (46 months) (*p* = 0.006) or MNA cases (22 months) (*p* = 0.0034). In the study cohort of 77 NB patients, 30 were males and 47 were females. Of the 21 patients with *ATRX* aberrations, 13 were females and 8 were males, while among the 23 patients with *TERT* SV, 13 were females and 10 males (Table 1).

### 3.7. Gene Expression Analysis with Survival Correlation

Gene expression analysis in non-*MYCN*-amplified NB tumors showed that higher expression levels of *SPRTN*, *FH*, *EXO1,* and *SMYD3* were associated with worse overall survival in both the Versteeg and SEQC cohorts, where only *EXO1* displayed Bonferroni-corrected statistical significance in both. Low expression of *TRIM67* was associated with worse overall survival but as *SPRTN*, *FH,* and *SMYD3*, only presenting corrected statistical significance in the SEQC cohort (Appendix A).

## 4. Discussion

High-risk NB tumors can be subclassified based on genetic aberrations, where 11q-deletion and MNA mark two distinct groups with respect to biology and genetics. NB patients with 11q-deleted tumors are generally older at the time of diagnosis and show a slower disease course than MNA tumors [4]. Additional indications of dismal outcomes are the presence of RAS or p53 pathway mutations or the presence of active TMM, such as MNA, *TERT* SV, or *ATRX* aberration [5,34]. In our cohort, a statistical difference in the median age at diagnosis was seen comparing the subgroups hallmarked by *ATRX* aberration, *TERT* SV, or MNA, with *ATRX*-aberrated cases (97 months) being twice as old at onset as *TERT*-rearranged cases (46 months) and four-times as old as MNA cases (22 months). This agrees with other studies that typically identified *ATRX* aberrations in patients older than 5 years of age [35,36].

Also consistent with a previous study [32], we found that *ATRX* aberrations and *TERT* SV are predominantly found in 11q-deleted tumors. Among the 21 unique patients with tumors positive for *ATRX* aberrations, we could perform copy number profiling for 19 cases, showing that 14 (74%) presented an 11q-deletion. The remaining five cases with *ATRX* aberrations had a profile representative of 17q-gained (n = 2), other structural (n = 2), and MNA (n = 1). A study by van Gerven et al. indicated that 11q-deletions are enriched in NB tumors with *ATRX* deletions but not *ATRX* point mutations [32]. However, in our study, all three patients with SNVs in *ATRX* also had an 11q-deletion. The *ATRX* deletions were recurrently affecting exon 2-10 (n = 4) and exon 2-9 (n = 2), both associated with in-frame fusions, with the latter due to exon 10 skipping [5,32,33]. An additional four *ATRX* deletions were also expected to be in frame: deletion of exon 3–9, exon 2–13, exon 7–12, and exon 20–25, respectively (Table 2). Whereas the impact of the exon 20–25 deletion is unclear due to the unknown function of the corresponding protein segment, all other in-frame deletions are expected to cause loss of the EZH2-interacting domain and the nuclear localization signal as well as perturbation or complete loss of the DAXX domain. To date there are no reports of differences in clinical presentation or outcome for patients with in-frame deletions versus patients with nonsense or missense mutations in *ATRX* [32,37,38]. In this study, four patients with *ATRX* multi-exon deletions died of the disease; none of them were predicted as in frame. Eight patients with *ATRX* aberration survived 5 years with a mix of in-frame and out-of-frame deletions within this group. As the follow-up times were limited for some patients in this cohort, correlation to survival in relation to the presence of active TMM mechanisms was difficult to achieve. All investigated tumors in this study with *ATRX* structural aberrations were scored as positive for the presence of c-circles (Figure 3), in agreement with previous studies [14,15]. The presence of ALT despite the lack of obvious *ATRX* aberrations was indicated, and Koneru et al. reported that 40% of ALT tumors in their study lacked genomic alterations in known ALT-associated genes [14]. Among our cases without established presence of *TERT* or *ATRX* aberrations or MNA, none were c-circle positive, although four cases with 11q-deletion showed strong signals in both Phi29+ and Phi29− (Figure 3). Case 66R3C, with strong signal in both Phi29+ and Phi29−, displayed long telomeres in the tumor and presented a confident score of 0.97 for methylation subclass “ALT/TERT TMM positive”, thus supporting the presence of active TMM. Whether this is related to ALT, to the so-called ever-shortening telomeres [13], or represents a novel TMM warrants further investigation.

Telomerase expression can be up-regulated in NB tumors via multiple genetic and epigenetic mechanisms, including amplifications, super-enhancer hijacking, promoter hypermethylation, or through *MYCN*-driven re-expression [5,39]. Among the 50 sequenced cases included in this study, no *TERT* promoter mutations were detected, in accordance with a previous report [7]. Instead, *TERT* re-arrangement was found in 24 samples from 23 patients in our study. For cases analyzed with WGS, the translocation partner of *TERT* was identified and indicated multiple loci with established functions in NB tumorigenesis, such as *MYCN* (2p), *HAND2* (4q), *GATA3* (10p), *CCND1* (11q), *RBFOX1* (16p), and *DLG2*, a gene with a role in neural differentiation and a tumor suppressor gene candidate in 11q-deleted NB [40,41] (Figure 1A). The majority of these fusions placed *TERT* in close proximity to previously identified super-enhancer elements, commonly located within 300 kb of the fusion point (Figure 1B and Appendix A). These alterations will likely cause elevated *TERT* expression, but additional analyses of gene expression would be required to validate to what extent *TERT* levels are affected by the different fusions. The co-occurrence of MNA and *TERT* SV was seen in five cases, wherein three also presented a the 11q-deletion. This indicates that two different means for *TERT* activation are present in a subset of tumors. However, given the possible presence of heterogeneous MNA where neoplastic cells with and without MNA co-exist within the tumor, the limitation of the used methods does not allow us to deduce if MNA and *TERT* are present in the same cell population.

MNA together with *ATRX* deletion has been reported to be incompatible in NB due to the replicative stress caused by defects in the ATRX–histone chaperone complex together with the stress induced by MYCN-mediated metabolic reprogramming [42]. In agreement with this, several studies have reported MNA and *ATRX* aberration to be mutually exclusive in NB [37,43,44]. However, in our study, one case (16R4) displayed co-occurrence of *ATRX* aberration and MNA. The *ATRX* exon 20--25 deletion seen in this tumor from a patient diagnosed at a late age (84 months) is expected to be in frame but is somewhat different from the other implicated in-frame fusions in this study, given a more C-terminal location with retention of the nuclear localization signal and DAXX domain. Although no particular function has been recognized for the protein segment corresponding to exon 20–25, the sample was indeed c-circle positive (Figure 3). Classification through methylation gave the highest score (0.68) to subclass “ALT/TERT TMM positive”, although subclass “MYCN type” received a score of 0.29, indicating a mixed ALT and *MYCN* phenotype. Heterogeneous tumors with amplified *MYCN* and the presence of ALT markers seem to occur but are very rare [15,42]. As clonal heterogeneity is common in NB, the manifestation of ALT and MNA could represent different subpopulations and, also, in this sample, the used methods are limited to deduce if MNA and *ATRX* aberration are present in the same cell population.

Investigation of segmental alterations on chromosome 1 among NB tumors showed a strong predilection for distal 1q-deletion in tumors with *ATRX* aberration (12/21, 57%), as compared to other tumors of other genomic profiles. Distal 1q-terminal deletion was not present in any of the other tumors, regardless of genomic subtype, except in one 11q-deleted case 73R5D (Figure 5B). This also agrees with a recent study that reports association between 1q-terminal loss and ALT [15]. Alteration of *ATRX* alone is not sufficient to induce ALT [22,45] and, thus, other cellular events must be necessary but are so far elusive. A parsimonious interpretation would be that an ALT suppressor is located within the 1q region.

Genes within the 1q commonly deleted region and a functional role in genome stability and/or chromatin accessibility include *SPRTN*, *EXO1, FH,* and *SMYD3* (Appendix A). The *FH* gene, previously suggested as a tumor-suppressor gene candidate in 1q-deleted NB [46], encodes fumarate hydratase and is involved in non-homologous end-joining repair, promoting DNA repair through the inhibition of histone H3 demethylation [47]. *SMYD3* is a lysine methyltransferase, for which the catalytic products include H3K4me3 as well as H4K20me3. H4K20me3 is enriched in heterochromatin regions including the telomeres and has been shown to be crucial in sustaining telomere integrity [48]. Disruption of the *SPRTN* gene causes heterogenous telomere size [49], possibly due to the ineffective release of stalled replication forks at difficult-to-replicate regions such as the telomeres [50]. The exonuclease *EXO1* also has a role in the rescue of stalled forks and DNA repair but also in the formation of the single-stranded 3`overhang needed for t-loop generation at telomere ends [51]. Hartlieb et al. proposed *EXO1* as an ALT suppressor candidate gene based on reduced gene expression in all ALT-positive tumors [15]. Gene expression analysis showed that high expression levels of *EXO1* as well as *SPRTN*, *FH,* and *SMYD3* were associated with worse overall outcomes in non-MNA patients (Appendix A).

Located directly distally of the chr1q42.2 breakpoint is *TRIM67*. Although without an established role in the context of ALT, *TRIM67* has been indicated as a tumor suppressor gene in other malignancies [52,53]. TRIM67 has been shown to induce neural differentiation through the regulation of Ras signaling and to attenuate cell proliferation and enhance neurogenesis when overexpressed in an NB cell line [54]. Gene expression analysis showed that low expression of *TRIM67* was associated with worse overall outcomes in non-MNA patients (Appendix A). Further studies are needed to determine whether *TRIM67* has a role in ALT, but the focus of breakpoints just proximal of this gene is intriguing.

While there was an obvious lack of 1q-deletions in *TERT*-rearranged cases, we observed a preference for 1q-gain, present in almost half of the cases with *TERT* SV (11/23, 48%). Among other subgroups of NB tumors, distal 1q-gain was seen in only 2/21 (9.5%) among cases with *ATRX* aberration and in 2/15 (13.3%) tumors with 11q-deletion but without genomic alteration of *TERT* or *ATRX*. In contrast to the hot-spot region associated with 1q-deletion, the break points for 1q-gain were dispersed over a large region, which limits the possibility to pinpoint a specific candidate gene.

Our study underscores the distinctive molecular landscape of high-risk NB, revealing that *ATRX* aberrations and juxtaposition of *TERT* are predominantly associated with the 11q-deleted NB subtype. These findings are important for understanding the etiology and molecular pathogenesis of NB and further support that high-risk NB is propagated by different molecular alterations affecting telomere maintenance and immortalization (Figure 6). This merits further investigation into the therapeutic strategies directed against the telomerase or ALT pathway. Targeting telomerase-activated NB with MNA or the type of *TERT* SVs described in this study is an attractive concept, although the full promise of telomerase targeting in malignant disease has not yet been realized. This is because the clinical trials of the telomerase inhibitor Imetelstat have been halted due to unacceptable toxicity [55,56]. However, other novel means for telomerase or telomere targeting show promising results [57,58,59]. Exploitation of the genetic instability and DNA damage response often observed in ALT cells could be essential for targeting these tumors. Thus, potential therapeutic strategies for targeting the ALT phenotype have been reported with inhibition of PARP, ATM, or ATR showing selective sensitivity in NB cell lines with *ATRX* aberrations [60,61,62]. However, the approach to therapeutic targeting of ALT is still elusive due to the complexity and the yet-unknown factors involved in the ALT mechanism. For instance, whether the *ATRX*-negative NB tumors with strong signals in both Phi29+ and Phi29− (Figure 3) also define an ALT group that could benefit from therapeutic strategies targeting genomic instability and/or DNA repair is not clear and needs to be addressed in future work. Also, if the frequent 1q-deletion seen in *ATRX*-deleted cases (Figure 5) indeed is an enabler of ALT, genes within in this region could be exploited for therapeutic vulnerabilities. Another caveat in TMM targeting is the heterogenous nature of NB, with cases presenting the co-occurrence of MNA and ALT or MNA and *TERT* SV. If the different TMMs represent divergent cell populations, one single TMM-targeting strategy might not be sufficient. Further studies on potential combination therapies that include the mitigation of TMM will be pivotal in order to increase the survival rate of high-risk NB in general and especially the clinically challenging group with 11q-deletions.

## 5. Conclusions

Our investigation of telomere maintenance mechanisms in a Swedish NB cohort reveals that *ATRX* aberrations and the juxtaposition of *TERT* are predominantly associated with the 11q-deleted NB subtype and that *ATRX*-mutated tumors display a strong predilection for 1q-deletion. C-circle analysis shows a strong correlation between *ATRX* and an ALT phenotype but also the presence of a subgroup of NB with very extensive telomeres, despite the lack of an *ATRX* aberration. Given the importance of TMM in high-risk NB, the development of new therapeutic strategies targeting these mechanisms is urgent to improve outcomes in this vulnerable group of patients.

## Figures and Tables

**Figure 1 cancers-15-05732-f001:**
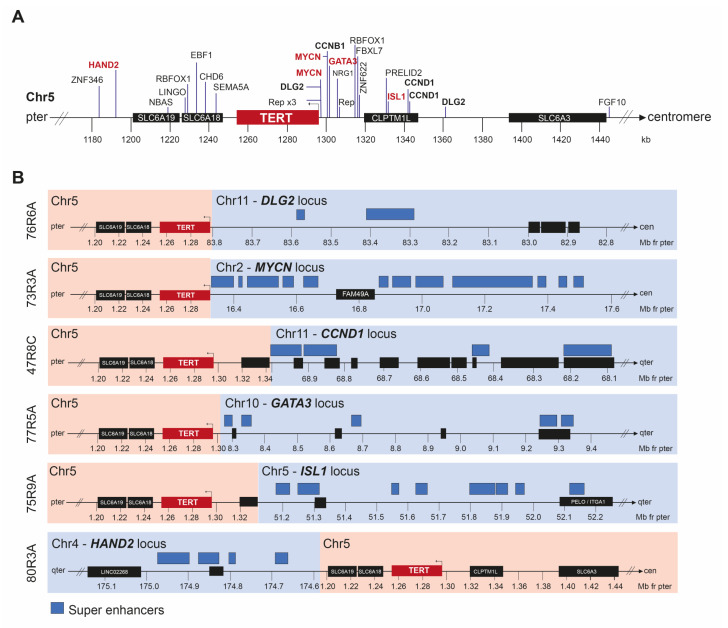
*TERT*-associated alterations. (**A**) An overview of the location of breakpoints affecting the *TERT* locus together with corresponding gene regions associated with the translocation. (**B**) Detailed view showing the juxtaposition of *TERT* in relation to super enhancers for six different cases. Super enhancers are specified by blue boxes with positions as indicated by previous studies by van Groningen et al. 2017 [29] and Gartlgruber et al. 2021 [30].

**Figure 2 cancers-15-05732-f002:**
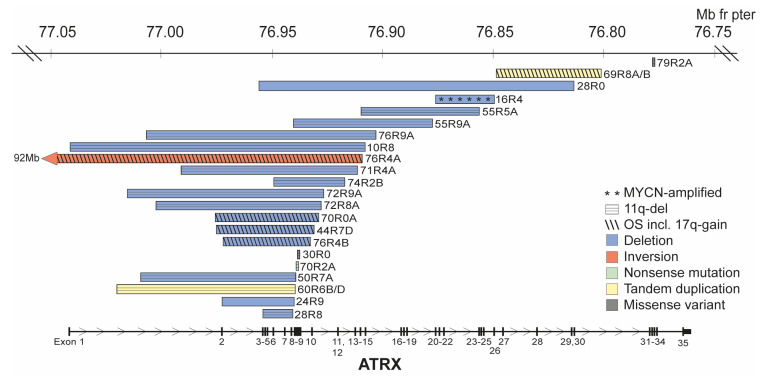
*ATRX*-associated alterations. Visualization of position and size of detected *ATRX* deletions together with annotation of genomic subgroup.

**Figure 3 cancers-15-05732-f003:**
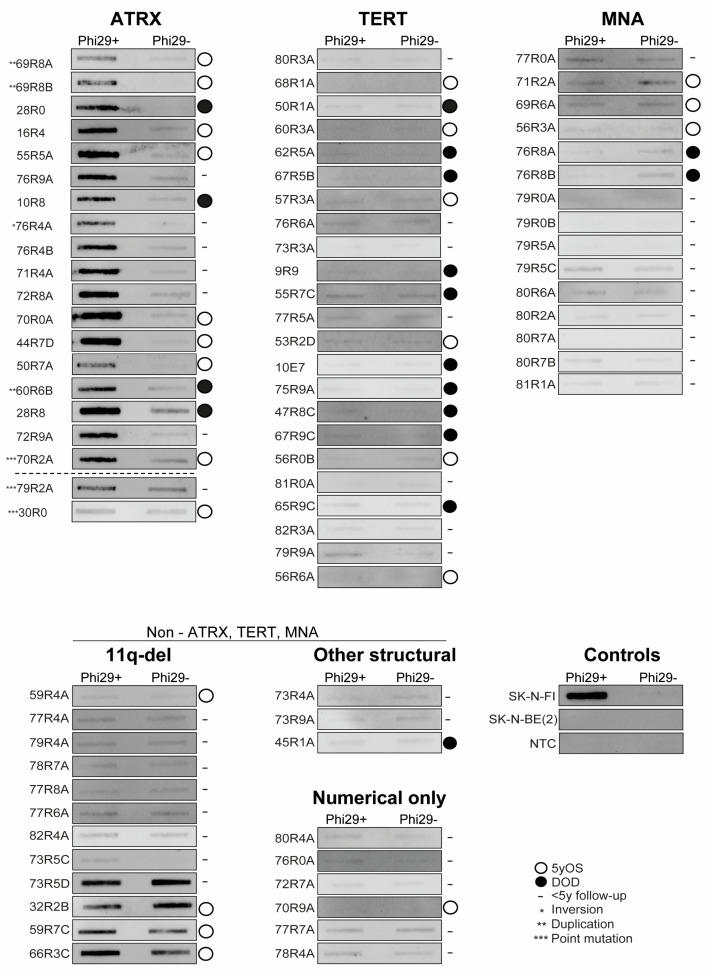
C-circle assay. C-circle assay slot blots are shown for all samples subdivided in groups according to presence of genomic alterations in *ATRX*, *TERT,* or MNA as well as a reference group consisting of NB samples lacking these aberrations. Cases in the *ATRX* group with other alterations than multi-exon deletions are indicated by asterisks.

**Figure 4 cancers-15-05732-f004:**
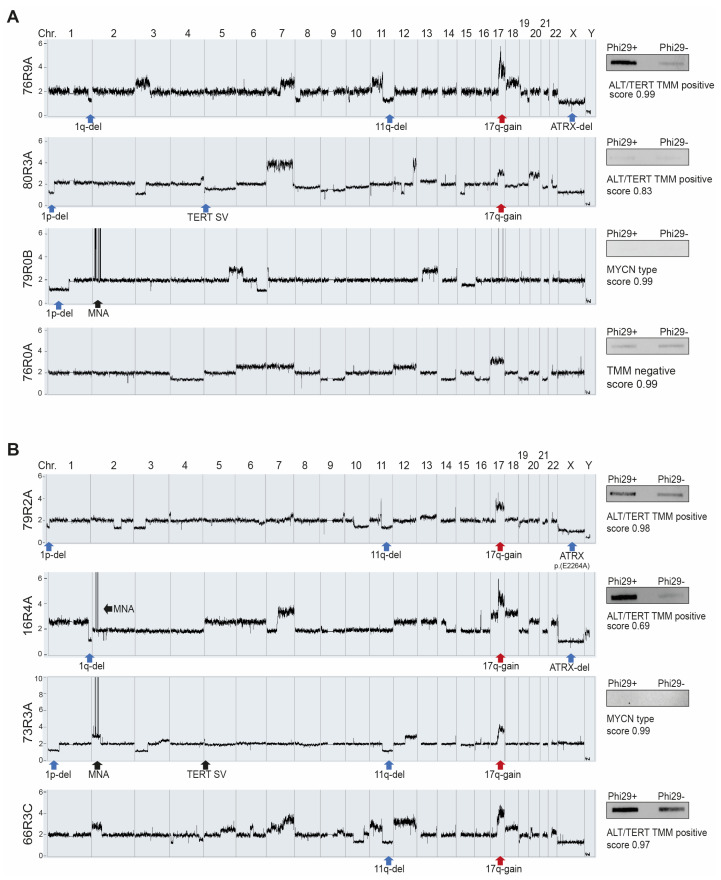
Common cases and cases of particular interest. Genomic tumor profiles generated by SNP microarray are shown as genome-wide chromatograms and aberrations are indicated with blue arrows for loss, red arrows for gain and black arrows for amplification. C-circle assay results by slot blot are shown to the right of respective chromatogram together with scores and neuroblastoma subclass indicated from methylation-based classification. (**A**) NB cases representative of different genomic subtypes. (**B**) NB cases with indefinite features.

**Figure 5 cancers-15-05732-f005:**
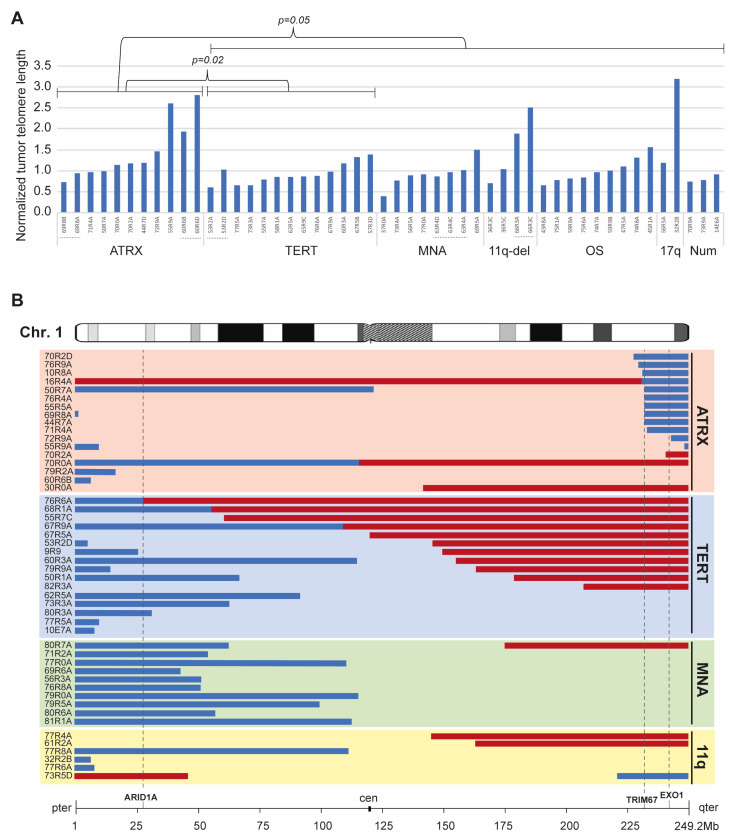
Telomere length estimation and segmental aberrations at chromosome 1. (**A**) Normalized tumor telomere length as calculated from WGS data using TelSeq software v.1.0 are shown as blue bars. Genomic subgroup as indicated on the X axis; presence of alteration in *ATRX*, *TERT,* MNA or 11q-deletion (11q-del), other segmental alterations (OS), 17q-gain (17q) and numerical only (Num). Different tumor materials stemming from same individual are marked by dotted line. (**B**) The samples with chromosome 1 aberrations are grouped into four categories according to alterations in *ATRX*, *TERT*, MNA and 11q-deleted (non- ATRX, TERT, MNA). Blue bars represent deletions and red bars represent gains. Genes of interest are highlighted with a dashed line.

**Figure 6 cancers-15-05732-f006:**
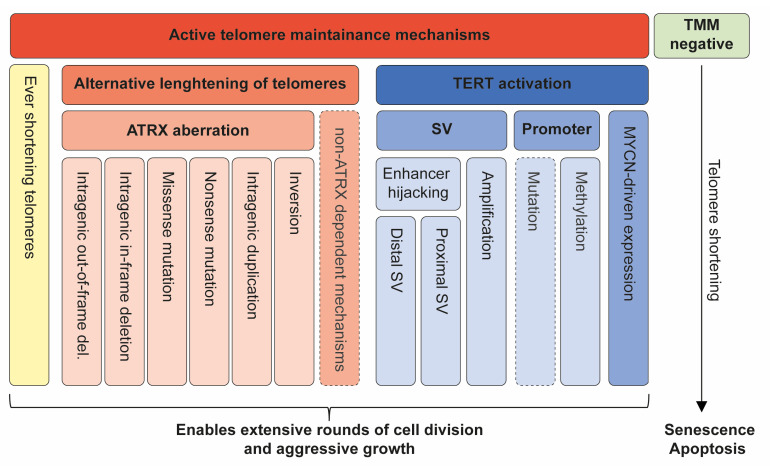
Overview of TMM in neuroblastoma. Immortalization is a hallmark of tumor progression and can be achieved through different molecular mechanisms commonly connected to an *ATRX* associated ALT phenotype or telomerase activation.

**Table 1 cancers-15-05732-t001:** Demographic and genomic features of the study cohort.

Characteristics	Study Cohort	ATRX Aberration	TERT SV
Cohort size	77	21	23
**Gender**			
Female	47	13	13
Male	30	8	10
**Age (months)**			
Median/Average age at diagnosis months (range)	36/53 (2–290)	80/97 (33–290)	36/47 (15–158)
**Outcome**			
DOD	16	4	9
NED	10	3	3
AWD	17	5	6
ND	34	9	5
**Genomic subgroup ***			
MNA	14	1	2
MNA + 11q-deleted	3	0	3
11q-deleted	41	14	14
17q-gain/Numerical only/Other structural	17	4	4
ND	2	2	0
**Genomic alterations**			
ALK-mutation/amplification	16	4	2
1p-del	32	6	13
1p-gain	2	1	0
1q-del	13	12	0
1q-gain	17	3	11

Abbreviations: MNA, MYCN amplified; DOD, dead of disease; NED, no evidence of disease; AWD, alive with disease; wcg, whole chromosome gain; SV, structural variant; ND, not determined. * Genomic subgroup given according to Carén et al. 2010 [4].

**Table 2 cancers-15-05732-t002:** Details of ATRX-associated alterations.

Case-ID	Method	Gender	Aad Months	Position (hg19)	Alteration Type	Affected Exons	In-Frame Fusion
10R8	MLPA	Female	83	chrX:76,907,604-77,041,702	Del	Del: exon 1–exon 15	Translation start loss?
76R4B	WGS	Female	222	chrX:76,932,561-76,972,164	Del	Del: exon 3–exon 9	Yes
24R9	MLPA	Male	290	chrX:76,937,012-76,972,720	Del	Del: exon 2–exon 9	Yes *
50R7A	WGS	Female	92	chrX:76,939,168-77,009,521	Del	Del: exon 2–partial exon 9	Yes *
72R9A	WGS	Male	52	chrX:76,926,307-77,015,566	Del	Del: exon 2–exon 10	Yes
72R8A	Array	Female	37	chrX:76,927,687-77,001,900	Del	Del: exon 2–exon 10	Yes
70R0A	WGS	Male	101	chrX:76,928,798-76,975,900	Del	Del: exon 2–exon 10	Yes
44R7D	WGS	Female	33	chrX:76,931,016-76,975,296	Del	Del: exon 2–exon 10	Yes
71R4A	WGS	Female	123	chrX:76,911,637-76,991,559	Del	Del: exon 2–exon 13	Yes
76R9A	Array	Male	59	chrX:76,902,858-77,006,971	Del	Del: exon 2–exon 15	No
28R0	MLPA	Male	166	chrX:76,812,922-76,972,720	Del	Del: exon 2–exon 30	No
28R8	MLPA	Male	28	chrX:76,954,117-76,940,431	Del	Del: exon 3–exon 8	No
74R2B	WGS	Female	42	chrX:76,917,132-76,949,209	Del	Del: exon 7–exon 12	Yes
55R9A	WGS	Female	55	chrX:76,877,184-76,940,168	Del	Del: exon 9–exon 19	No
55R5A	Array	Female	41	chrX:76,856,156-76,909,674	Del	Del: partial exon 14–exon 22	No
16R4	Array	Male	80	chrX:76,850,122-76,881,028	Del	Del: exon 20–exon 25	Yes
76R4A	WGS	Female	222	chrX:76,909,074-94,210,093	Inv	Break: intron 14	NA
69R8A/B	WGS	Female	117	chrX:76,800,300-76,847,812	Dup	Dup: exon 27–exon 30	No
60R6B/D	WGS	Female	50	chrX:76,939,370-77,020,418	Dup	Dup: exon 2–partial exon 9	NA
70R2A	WGS	Female	266	chrX:76,938,326	SNV Nonsense	Exon 9 NM_000489.6: c.2422C>T p.(R808*)	NA
79R2A	WGS	Male	80	chrX:76,778,788	SNV Missense	Exon 31 NM_000489.6: c.6791A>C p.(E2264A)	NA
30R0	WGS	Female	29	chrX:76,937,825	SNV Missense rs200709847	Exon 9 NM_000489.6: c.2923G>A p.(D975N)	NA

Abbreviations: WGS, whole genome sequencing; Aad, Age at diagnosis; Dup, duplication; Del, deletion; Inv, inversion; SNV, single nucleotide variant; * in-frame with exon 10 skipping as suggested by Ackermann et al., 2018 [5].

## Data Availability

The data presented in this study are available in this article and Appendix A.

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
