# Peer review of "Telomere Maintenance Mechanisms in a Cohort of High-Risk Neuroblastoma Tumors and Its Relation to Genomic Variants in the TERT and ATRX Genes"

_cancers, 2023, doi:10.3390/cancers15245732_

Round 1
Reviewer 1 Report
Comments and Suggestions for Authors
In their work, the authors discuss the connection between microdeletion and known mutations, such as TERT, in neuroblastomas. Overall, the manuscript is very well written, clearly structured and easy to understand. The results provide valuable information for future studies.
I have only minor comments:
The reader may be somewhat confused about the exact size of the study cohort:
- Page 3, line 111: tumour sampes n=232
Table 1, Study cohort - but study cohort only includes column 1? It is not entirely clear here whether columns 2 and 3 belong to the study cohort (subgroup) or are a comparison group
Page 7, section 3.3 - 22 patients with ATRX (Table 1, however, 21?)
Please check the numbers in the text. An overview of the breakdown of the cohort and the groups with ARTX and TERT mutation may help with understanding
- Page 2, line 55: dot after "pairs." please remove
- Page 4, line 164 - 200 ul - please correct unit
Reviewer 2 Report
Comments and Suggestions for Authors
This is a very nice evaluation of the role of telomere maintenance mechanisms in high risk neuroblastoma among a Swedish patient cohort. The data presented is similar to other data evaluating TMMs.
How is the presented data unique? There is a sentence at the end of the paper suggesting that therapies that target the difference subgroups need to be determined. This is a start to addressing how this work can be used.
To make this paper more meaningful, there needs to be focus on how the findings add to the current understanding of TMMs and how this can be used to optimize therapeutic strategies.
Reviewer 3 Report
Comments and Suggestions for Authors
Seems no need for section 2.7 Kaplan-Meyer analyses of overall survval - none are presented in Results
Comments on the Quality of English Languagesatisfactory quality, only a few minor edits can be considered
